# Guided Filter Regularization for Improved Disentanglement of Shape and Appearance in Diffeomorphic Autoencoders

**Uzunova H** [1]                                                    HRISTINA.UZUNOVA@DFKI.DE

**Handels H** [1,2]                                                  HANDELS@IMI.UNI-LUEBECK.DE

**Ehrhardt J** [2]                                                   EHRHARDT@IMI.UNI-LUEBECK.DE

[1] *German Research Center for Artificial Intelligence, Lübeck, Germany*

[2] *Institute of Medical Informatics, University of Lübeck, Lübeck, Germany*

## Abstract

Diffeomorphic and deforming autoencoders have been recently explored in the field of medical imaging for appearance and shape disentanglement. Both models are based on the deformable template paradigm, however they show different weaknesses for the representation of medical images. Diffeomorphic autoencoders only consider spatial deformations, whereas deforming autoencoders also regard changes in the appearance, however no uniform template is generated for the whole training dataset, and the appearance is modeled depending on a very few parameters. In this work, we propose a method that represents images based on a global template, where next to the spatial displacement, the appearance is modeled as the pixel-wise intensity difference to the unified template. To however ensure that the generated appearance offsets adhere to the template shape, a guided filter smoothing of the appearance map is integrated into an end-to-end training process. This regularization significantly improves the disentanglement of shape and appearance and thus enables multi-modal image modeling. Furthermore, the generated templates are crisper and the registration accuracy improves. Our experiments also show applications of the proposed approach in the field of automatic population analysis.

**Keywords:** Disentanglement, Diffeomorphic autoencoder, Guided filter

## 1. Introduction

The disentanglement of shape and appearance is a prominent computer vision task (Yang et al., 2020; Lorenz et al., 2019; Ding et al., 2020) and it becomes more and more relevant for the medical image analysis (Liu et al., 2020; Wilms et al., 2017). In the medical field, multiple devices, imaging techniques or parameters are often applied for the imaging of the same anatomical structures. Since the appearance of such acquisitions strongly varies, it is important to distinguish between changes in the anatomical shapes and changes in the intensity profile. Statistical shape and appearance models enable separate representations of the shape and appearance, e.g. (Wilms et al., 2017), however they typically require preprocessing to extract corresponding point clouds for shape information. Deep learning approaches like (Liu et al., 2020; Uzunova et al., 2020) partly establish a shape and appearance disentanglement by proposing topology-constrained appearance domain translation using conditional generative adversarial networks. However, those methods are not able to extract the shape by themselves, and thus the simple topology constraints like rough labels or image edges do not guarantee an accurate disentanglement. Another possibility is

a disentanglement directly in the latent space as in (Ding et al., 2020), however, shape and appearance are not explicitly modeled.

Another type of methods assume that images can be represented as deformed versions of a given template. For example, diffeomorphic autoencoders (Bône et al., 2019) model images as spatial displacement offsets to a dynamically generated template, however do not consider any occurring appearance changes. A similar approach from (Shu et al., 2018) generates an appearance template and a displacement field for each given image. Yet, a major drawback of this method is the lack of a global template for the given dataset, which interferes with the reliability of the disentanglement.

In this work, we propose to model both the shape and appearance of medical images as spatial and intensity offsets to a global template in an autoencoder network. This enables the group-wise registration of images with different intensity characteristics, e.g. multi-modal images. A related approach was developed in parallel by (Bône et al., 2020) in the context of image metamorphoses (Trouvé and Younes, 2005) and intended for the registration of images with pathologies. We further propose an appearance regularization by integrating guided filtering (He et al., 2013) in the network to ensure that the appearance offsets are guided by the structures of the template. Our experiments show that the guided filtering leads to an improved disentanglement of shape and appearance, generates sharper and more precise templates and improves registration accuracy. Furthermore, group-wise registration of images from scanners with different intensity characteristics and even multi-modal images without using dedicated metrics like mutual information is possible. We also show, that our approach enables automatic population analysis of intensity characteristics even for small brain structures.

## 2. Methods

The deforming autoencoder presented by (Shu et al., 2018) interprets an image $X$ as a composition of two parts: a deformation-free appearance template $T$ and a deformation field $\varphi$, such that $X \approx T \circ \varphi$, with $\circ$ denoting the warping function (Figure 1). The appearance and deformation field are generated by an autoencoder architecture with two separate decoders. The two parts of the composite latent vector $Z = [Z_T, Z_\varphi]$ generated by the encoder is fed into the appearance and deformation decoders respectively. The objective

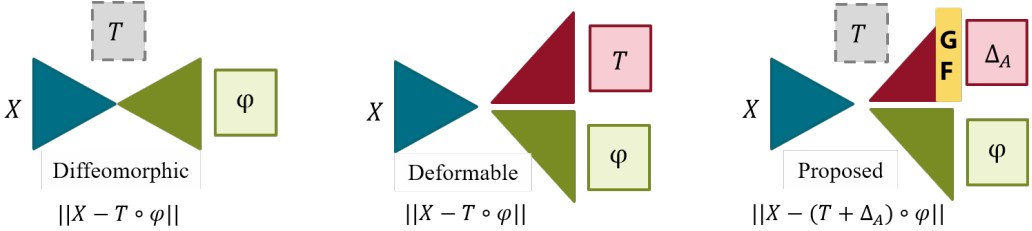

Figure 1: Diffeomorphic (Bône et al., 2019), deformable (Shu et al., 2018) and proposed autoencoder with guided filtering (GF) layer. Training objectives below: input $X$, template $T$, displacement field $\varphi$, appearance offset map $\Delta_A$.

of the training enforces the warped template to be as similar to the input image as possible. Although shown to succeed for natural images, this approach has several disadvantages: The deformation-free appearance image is generated for each image separately, meaning that no reliable mapping to the same image space is possible. Moreover, as the authors indicate, in order to achieve a reliable disentanglement, the dimension of $Z_T$ needs to be considerably small and therefore must be carefully adjusted (Siebert and Heinrich, 2020).

The problem of generating a global template for each input image is tackled in (Bône et al., 2019). The authors of this work use an autoencoder to extract a deformation field $\varphi$ from an input image $X$ such that $X \approx T \circ \varphi$, where $T$ is a global parameterized template that is generated by joint optimization during the back-propagation process (Figure 1). A main drawback of this approach is, that yet only spatial deformations of the inputs are considered, and intensity deviations from the template are not modeled.

Motivated by the above issues, we propose a method that considers both shape and appearance, but uses a single global template. Furthermore, a guided filter regularization for the appearances ensures a more robust disentanglement of shape and appearance.

## 2.1. Joint Appearance and Shape Autoencoder

For the proposed approach, each image $X_i$ is represented as $X_i \approx (T + \Delta_{A_i}) \circ \varphi_i$, where $\Delta_{A_i}$ and $\varphi_i$ are image-specific and $T$ is global (Figure 1 and appendix). Here, $\varphi_i$ is a displacement field, that maps the template to the image space and $\Delta_{A_i}$ approximates the pixel-wise intensity difference $(T - X_i \circ \varphi_i^{-1})$. We also refer to $\Delta_{A_i}$ as "appearance map".

Similarly to (Shu et al., 2018) the encoder generates a composite latent representation $Z = [Z_T, Z_\varphi]$ and two decoders each generate an appearance map and a shape displacement. The appearance map is added to the dynamically generated template and then warped with the displacement field originating from the displacement decoder. Like in (Bône et al., 2019), a global template is implicitly learned during the back-propagation process. However, the straight-forward generation of the appearance map does not guarantee its adherence to the template structures. Thus new structures can be generated or changed in a manner that imitates spatial deformations. Hence, the appearance map may also modify the shape of structures, i.e. the disentanglement of shape and appearance is not guaranteed. To cope with this problem, we propose a guided filter regularization approach for the appearance map in order to ensure its guidance by the template.

## 2.2. Guided Filter for Appearance Regularization

Guided filter (GF) (He et al., 2013) is an edge preserving image smoothing approach that considers a given guidance image $I$ for the smoothing of the input image $p$. The output $q_i$ at each pixel $i$ is assumed to be a linear transformation of $I$ in a window $\omega_k$ with radius $r$:

$$q_i = a_k I_i + b_k, i \in \omega_k. \tag{1}$$

$a_k$ and $b_k$ are window-specific linear coefficients that can be calculated in closed form from mean and variance of the guidance image $I$ in the window $\omega_k$ by minimizing $E(a_k, b_k) = \sum_{i \in \omega_k} \left( (a_k I_i + b_k - p_i)^2 + \epsilon a_k^2 \right)$ with regularization parameter $\epsilon$. The linear model in Eq.(1) ensures a guidance on $I$ since $q$ has an edge only if $I$ has an edge, i.e. $\nabla q_i = a_k \nabla I_i$.

As proposed in (Wu et al., 2018), GF can be implemented in a fully differentiable manner by calculating the mean and standard deviation values by using a box filter with radius $r$ and applying a linear model yielding $q$. Using this approach here, GF is implemented as the last layer of the appearance decoder in order to smooth the appearance map $\Delta_A$ guided by the generated template $T$. This ensures that the appearance map does not change the shape of the template, yielding better disentanglement. And, since the template is optimized during back-propagation, the GF also leads to a sharper and more representative template.

### 2.3. Implementation Details

The proposed architecture contains one linear and three convolutional layers in each encoder and decoder. In our experience, a comparably large $Z_T$ of size 64 delivered best results, whereas the size of $Z_\varphi$ is 512. Also, a Kullback-Leibler loss restricts the latent space to a normal distribution as typical for variational autoencoders (Kingma and Welling, 2014). An SSIM loss is used to ensure good image reconstructions (Zhou Wang et al., 2004).

In order to generate plausible deformation fields, multiple constraints are applied. First, diffeomorphism is enforced using a method based on static velocity fields (Arsigny et al., 2006). Thus, the displacement decoder generates a velocity field $v$ and the displacement field is calculated as $\varphi = \exp(v)$, where $exp(\cdot)$ can be approximated with the scaling-squaring algorithm and integrated into an end-to-end training process. Furthermore, the tissue deformations are enforced to be smooth and of small magnitude in order to learn average template shapes. Overall, the following objective can be formulated as follows:

$$\mathcal{L} = \mathcal{L}_{KL}(\mathcal{N}(0,1), \mathcal{N}(\mu_z, \sigma_z)) + \mathcal{L}_{SSIM}(X, (T + \Delta_A) \circ \varphi) + \alpha \underbrace{\sum_{j}^{d} ||\nabla v^{(j)}||_2^2}_{\text{diffusion reg.}} + \underbrace{\beta ||v||_1}_{\text{l1 reg.}}. \quad (2)$$

Here, $\mathcal{L}_{KL}$ denotes the Kullback-Leibler loss between the expected $\mathcal{N}(0,1)$ and the real latent distribution $\mathcal{N}(\mu_z, \sigma_z)$ with $\mu_z$ and $\sigma_z$ being the learned mean and standard deviation, respectively. $\mathcal{L}_{SSIM}$ is the SSIM loss between the input and its reconstruction and the last two terms serve the regularization of the velocity field weighted by $\alpha$ and $\beta$.

## 3. Experiments and Results

### 3.1. Data

**Brain MRIs:** For our experiments 577 MR brain images of normal adult healthy subjects from the IXI dataset[1] are used. For each MRI T1 and T2 sequences are considered. The images are acquired in three different hospitals and thus vary in appearance. Furthermore, there is demographic information available for each patient, including their age.

**Labeled brain MRIs:** For evaluation purposes, we use 30 subjects' T1 MRIs with ten selected labeled anatomical regions (Hammers et al., 2003)[1].

**SRI 24 Atlas:** For some of the experiments, an already existing atlas of the healthy adult brain from (Rohlfing et al., 2009) is used. All data is pre-processed using affine registration and all experiments are performed on extracted axial 2D slices of size $173 \times 211$.

---

1. www.brain-development.org

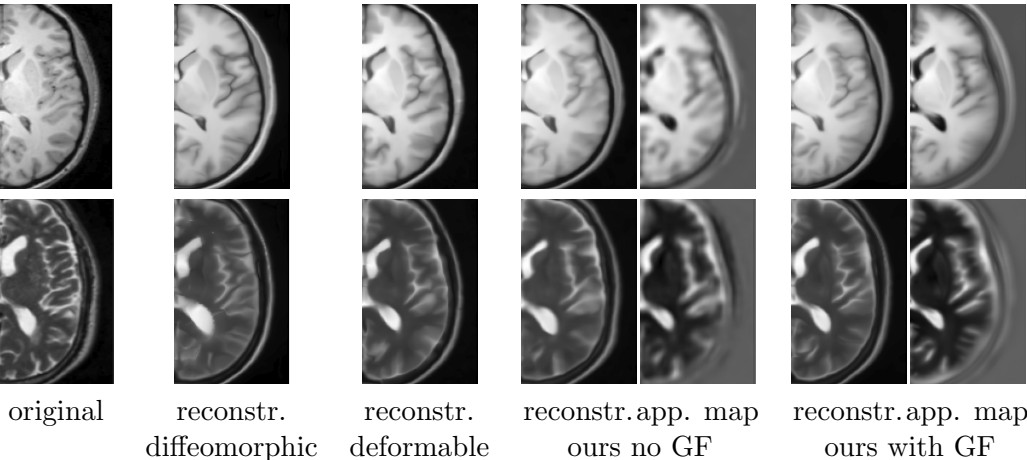

| original | reconstr. diffeomorphic | reconstr. deformable | reconstr. app. map ours no GF | reconstr. app. map ours with GF |

Figure 2: Reconstruction ability of all models and appearance maps when applicable. First row: T1 training only; second row: mixed modality training set.

## 3.2. Image Reconstruction Quality

To evaluate the images generated by the proposed model, the accuracy of reconstructed test images (generalization ability), and the specificity of randomly sampled images are measured using SSIM, MSE and MAE (see Table 3 in appendix B). The deformable autoencoder (Shu et al., 2018) and the diffeomorphic autoencoder (Bône et al., 2019) serve as baselines. Two training scenarios are examined: (1) training on T1 image sequences (single modality training), and (2) training on T1 and T2 sequences simultaneously (mixed modality training) in order to assess the shape and appearance disentanglement. Overall, all models yield comparable results (for additional results, see appendix). This is however expected, since the same objective is optimized and displacement and appearance can mutually compensate each other. Visually (Figure 2), the images reconstructed by the regularized model appear to be of a slightly improved quality.

Further, the generated templates of all models and training scenarios are qualitatively evaluated (Figure 3). Generally, the image-specific templates of the deformable autoencoder are blurry and inaccurate. In contrast, the diffeomorphic autoencoder generates a global template that is of good quality when trained on T1 images only, but is highly implausible for training with mixed modalities due to the lack of appearance variation modeling.

The proposed joint autoencoders generate templates of high quality where mixed datasets lead to templates representing a combination of T1 and T2 appearances. Still, for the mixed training set the non-regularized method yields less accurate results and artifacts can be observed in the ventricle regions. The proposed GF approach delivers sharp results with no visible artifacts regardless the training setup. Because the GF regularization impairs the generation of structures not available in the template, the network learns to represent relevant structures in the training set by edges in the template image. So, structures available only in a subset of images are still generated in the template and their presence is controlled by the additional map (Figure 4). The size of the latent space, the capacity of the decoder, and the contribution to the overall loss determine which structures are represented.

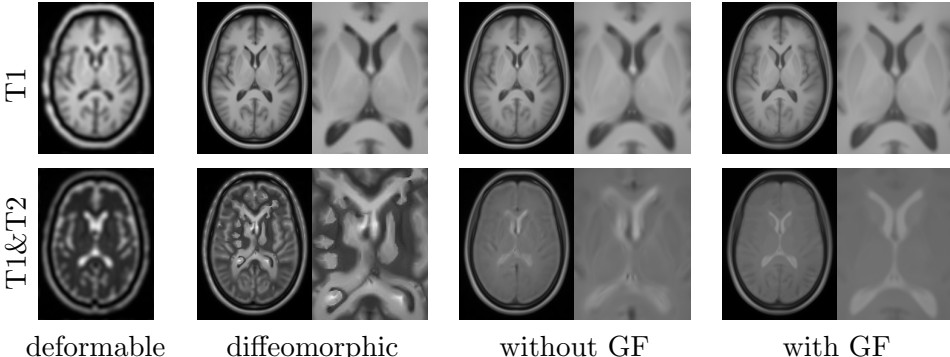

Figure 3: Template examples. Trained on T1 only (top) and mixed (bottom). Shown are: image-specific templates from (Shu et al., 2018); global templates and zoomed in areas from (Bône et al., 2019) and our method without and with GF.

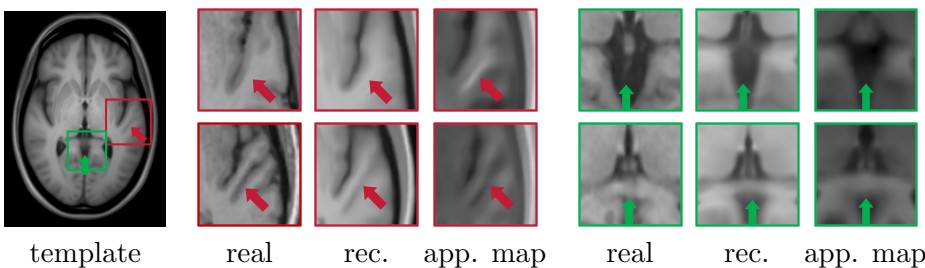

Figure 4: Structures not available in all images are present in the generated template and can be "turned on" or "turned off" by the appearance map.

## 3.3. Registration Accuracy

To evaluate the achieved registration accuracy of all models, three experiments are designed: First, the models are trained on the single and mixed modality datasets and tested on the 30 T1 images with available label maps. Thanks to the diffeomorphic formulation of the transformations, mean Dice values for the label maps $Y_i$ and $Y_j \circ \varphi_j \circ \varphi_i^{-1}$ can be computed in a pairwise manner, thus a total of 870 pairs is evaluated ($30 \times 29$). In a second experiment, the registration cycle consistency is assessed. The idea behind this experiment is that given an input image $X$ aligned to the templates $A$ and $B$ using $\varphi_{X \to A}$ and $\varphi_{X \to B}$, respectively, while $\varphi_{A \to B}$ maps $A$ to $B$, then $\varphi_{X \to A} \circ \varphi_{A \to B} \approx \varphi_{X \to B}$. For this experiment, the models are trained on the mixed dataset with two fixed templates (SRI atlas and an image from the dataset), i.e. no update of the template is performed during backpropagation. The cycle consistency is measured as the mean displacement error. Similarly, the last experiment assumes that when (rigidly aligned) T1 and T2 sequences of the same subject are registered to $A$, then $\varphi_{X_{T1} \to A} \approx \varphi_{X_{T2} \to A}$.

In all experiments, our approach with GF outperforms all other models. The registration accuracy of the proposed method is significantly improved (Table 1). Also, the cycle

Table 1: Registration results: Mean Dice for the pair-wise registration of 30 images. (*) indicates statistical significance compared to *ours with GF* in a two-tailed t-test.

| trained on | diffeomorphic | deformable | ours no GF | ours with GF |
|:---:|:---:|:---:|:---:|:---:|
| T1 | 0.76±0.08 | 0.69±0.11* | 0.74±0.10* | 0.76±0.10 |
| mixed | 0.64±0.11* | 0.63±0.11* | 0.67±0.09* | 0.69±0.09 |

consistency error with GF regularization drops to 5.5 pixels vs. 6.6 when no GF is applied. This tendency is followed up in the displacement consistency errors of T1 and T2 sequences: a mean displacement error of 7.4 is achieved with GF and 9.8 without using GF (see appendix).

### 3.4. Shape and Appearance Disentanglement

The previous experiments showed that the reconstruction ability of all models is comparable, however the registration accuracy of the GF approach is superior, leading to the conclusion that the proposed method enables improved shape and appearance disentanglement.

To further investigate this assertion, the latent vectors $Z_\varphi$ and $Z_T$ can be separately varied and decoded into the image space. In Figure 5, the results of the linear interpolation between the latent vectors of two images are shown. Please note, that when the appearance vector is varied, the shape vector stays fixed and vice versa. The results clearly show that the lack of GF regularization causes shape variations of the brain ventricles to be also captured by the appearance vector rather than the shape vector alone. When applying the proposed GF smoothing, the shape and appearance appear distinctly decoupled.

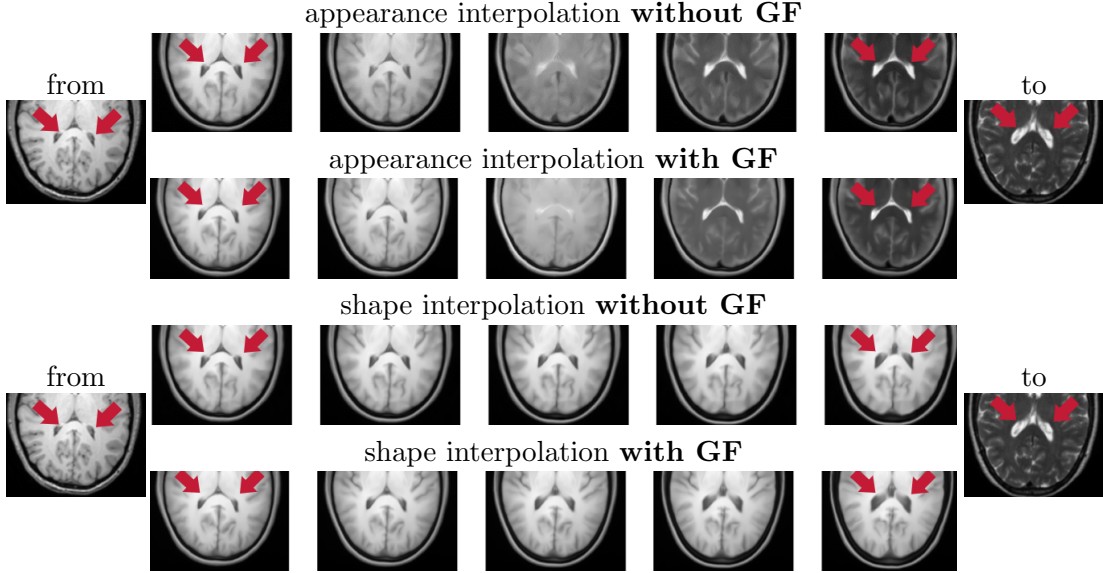

Figure 5: Visualization of the decoded images from interpolated latent vectors.

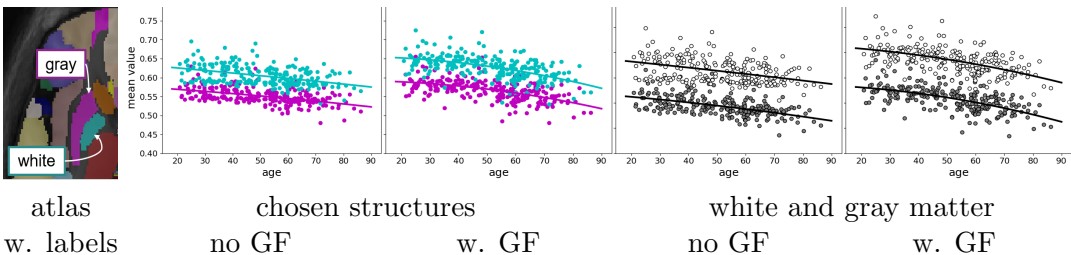

atlas            chosen structures            white and gray matter
w. labels        no GF            w. GF        no GF            w. GF

Figure 6: Age-related statistics. Plotted mean intensities of the two structures and white and gray matter w/o. and w. GF. The lines correspond to quadratic regression.

### 3.5. Example Application: Age-related Statistics

Several population-based studies involve the investigation of disease- or age-related effects on MR signal intensities for brain structures, e.g. the age-related decrease of MRI intensity of the gray and white matter (Ge et al., 2002; Lemaître et al., 2005; Tullo et al., 2019). This experiment demonstrates the applicability of our method for such studies even for small brain structures. Since we do not have access to quantitative MR sequences (Ge et al., 2002), 319 subjects acquired with the same MR scanner are selected to minimize device-dependent effects. The SRI24 atlas is used as a fixed template allowing for the usage of the given annotations of the gray and white tissue and anatomical structures. Two relatively small and closely located white and gray matter structures are chosen for close examination: left and right globus pallidus (cyan) and left and right putamens (magenta in Figure 3.5). By performing an atlas-based segmentation of the reconstructed test images and calculating the mean intensities over structures, age-dependent distributions are achieved. Since the GF approach improves registration, the discriminability of the two structures is enhanced (Figure 3.5). Regression analysis also reveals that the expected quadratic behavior of the age-intensity distribution is more prominent in the GF-based approach. Although different MR sequences are used, the regression curves are comparable to the results in (Ge et al., 2002) when the images are scaled to similar intensities (see appendix).

### 4. Conclusion and Discussion

In this work, we introduce a joint diffeomorphic autoencoder, where both spatial displacement and appearance offsets to a dynamically generated template are modeled simultaneously. To, however, enforce the appearance maps to conform to the template, we integrate a guided filter regularization into an end-to-end training. In our experiments, guided filtering improves the disentanglement of shape and appearance and the quality of the generated templates substantially. Moreover, the guided filter enhances mono- and multi-modal group-wise registration significantly, underlining the improved disentanglement. Also, the regularization method delivers a successful proof-of-concept in a scenario for the automatic population analysis of intensity characteristics even for small structures. Although our approach shows clear advantages in mono- and multi-modal settings, the guided filtering can suppresses the reconstruction of pathological structures, limiting its applicability in such scenarios. This will be a future research direction.

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

## Appendix A. Detailed Network Architecture

All used models share similar architectures with encoders containing each three convolutional layers with stride two and a two fully connected layer for the mapping to the mean and standard deviation vectors respectively. The decoders contain a fully-connected layer mapping the latent vector to the feature space and three convolutional layers combined with a bilinear upsampling layer that doubles the image size before each convolution. We use tangens hyperbolicus activation functions between all layers. For the deformable approach the size of the shape latent vector $Z_\varphi$ is set to 10 and the size of the appearance latent vector $Z_T$ to 512. For the diffeomorphic autoencoder, the size of the latent vector is 512, and for the proposed approach the size of $Z_\varphi$ is 64 and the size of $Z_T$ is 512. Those values were determined empirically, however note that the proposed method allows for a larger shape latent space. A schematic visualization of the proposed GF architecture is shown in Fig. 7. In all experiments, the networks are trained for 1000 epochs, however an early stopping strategy is used if the loss does not improve over ten epochs. Further, the batch size is set to 50 and an Adam optimizer with a learning rate of $1e^{-4}$ is used. The parameters $\alpha$ and $\beta$ for the loss function in Eq. 2 are both set to 10. All models are trained in a 4-fold-cross-validation manner and the measurements are averaged over all images and folds. The training code containing all preferred settings and used images is available at www.github.com/hristina-uzunova/GF_DAE.

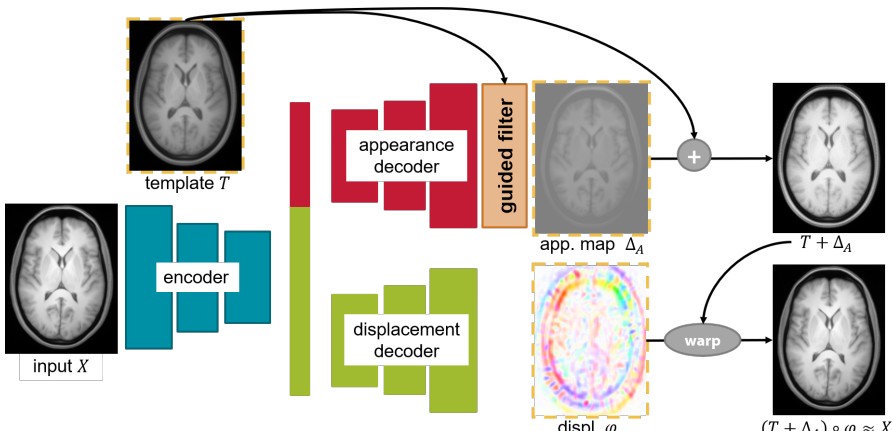

Figure 7: Overview of the proposed method. An input image is encoded into a joint latent space. One part of the latent vector is inputted into a displacement decoder, that outputs an image specific displacement $\varphi$ to the global template $T$. The other part of the latent vector is the input of an appearance decoder that generates a pixel-wise appearance offset map $\Delta_A$, such that $X \approx (T + \Delta_A) \circ \varphi$. The last layer of the appearance decoder is a guided filter layer that smooths the appearance offset following the guidance of the template. The dashed yellow borders indicate that the images are learned during backpropagation.

## Appendix B. Additional Experimental Results

Here, we show some additional experimental results for the experiments from Sec. 3.

The registration cycle consistency (Tab. 2) between two atlases is assessed as the mean displacement error $||\varphi_{X \to A} \circ \varphi_{A \to B} - \varphi_{X \to B}||$, where $X$ is an image aligned to the templates $A$ and $B$ using $\varphi_{X \to A}$ and $\varphi_{X \to B}$, respectively, while $\varphi_{A \to B}$ maps $A$ to $B$. The cycle consistency for T1 and T2 sequences is based on the similar assumption that when (rigidly aligned) T1 and T2 sequences of the same subject are registered to $A$, then $\varphi_{X_{T1} \to A} \approx \varphi_{X_{T2} \to A}$. Thus the mean distances $||\varphi_{X_{T1} \to A} - \varphi_{X_{T2} \to A}||$ are presented in Tab. 2.

| method | no GF | w. GF |
|---|---|---|
| two atlases | $6.39 \pm 0.61$ | $5.46 \pm 0.51^*$ |
| t1→t2 | $9.78 \pm 1.03$ | $7.36 \pm 0.60^*$ |

Table 2: Results of the cycle consistency and modality consistency experiment. The mean displacement errors for the registration to two different atlases and registration of T1 and T2 sequences to the same atlas as explained in Sec. 3. Without using guided filter (GF), significantly larger errors are achieved. Superscript $^*$ denotes statistical significance compared to *no GF* in a two tailed t-test.

In order to asses the quality of generative models, the commonly considered metrics specificity and generalization are also utilized here (Tab. 3). Specificity is the ability of a generative model to generate new realistic samples that are similar to the training data. Thus specificity is measured by generating the set of synthetic images $\mathcal{S}$ (here $|\mathcal{S}| = 100$) and calculating $\frac{1}{N_S} \sum_j^{N_S} \min_{\mathbf{r}_i \in \mathcal{R}} \{dist(\mathbf{r}_i, \mathbf{s}_j | \mathbf{r}_i \in \mathcal{R}, \mathbf{s}_i \in \mathcal{S}\}$ where $\mathcal{R}$ is a set of real images and $dist(\cdot, \cdot)$ a suitable distance metric. Generalization ability is the ability of the model to reconstruct unseen samples. Thus generalization can be calculated as $\frac{1}{N_R} \sum_i^{N_R} dist(\mathbf{r}_i, \widetilde{\mathbf{r}}_i)$, with $\widetilde{\mathbf{r}}_i$ being the reconstruction of the real image $\mathbf{r}_i \in \mathcal{R}$.

| training data | method | specificity | | | generalization | | |
|---|---|---|---|---|---|---|---|
| | | SSIM | MSE | MAE | SSIM | MSE | MAE |
| | deformable | 0.62 | 0.011 | 0.066 | 0.85 | 0.003 | 0.036 |
| T1 | diffeomorphic | 0.67 | 0.012 | 0.066 | 0.86 | 0.008 | 0.056 |
| | ours no GF | 0.69 | 0.011 | 0.065 | 0.88 | 0.003 | 0.029 |
| | ours w. GF | 0.68 | 0.013 | 0.068 | 0.87 | 0.003 | 0.032 |
| | deformable | 0.61 | 0.015 | 0.082 | 0.81 | 0.005 | 0.041 |
| T1&T2 | diffeomorphic | 0.61 | 0.017 | 0.084 | 0.85 | 0.007 | 0.051 |
| | ours no GF | 0.64 | 0.015 | 0.082 | 0.81 | 0.005 | 0.038 |
| | ours w. GF | 0.65 | 0.017 | 0.84 | 0.80 | 0.005 | 0.040 |

Table 3: Specificity and generalization abilityfor all training scenarios. Measured are structural similarity index (SSIM ↑) , mean squared error (MSE ↓) and mean absolute error (MAE ↓). Our method with and without guided filter (GF) is compared to the two baseline methods: diffeomorphic and deformable autoencoders.

| tissue | with guided filter–quadratic | $R^2$ | linear | $R^2$ |
|---|---|---|---|---|
| gray matter | $29 - 0.008 * \text{age} - 0.0005 * \text{age}^2$ | 37% | $30 - 0.054 * \text{age}$ | 35% |
| white matter | $34 - 0.002 * \text{age} - 0.0004 * \text{age}^2$ | 31% | $35 - 0.05 * \text{age}$ | 30% |
| putamen | $35 + 0.0007 * \text{age} - 0.0006 * \text{age}^2$ | 29% | $37 - 0.06 * \text{age}$ | 28% |
| pallidus | $39 + 0.0007 * \text{age} - 0.0006 * \text{age}^2$ | 30% | $41 - 0.006 * \text{age}$ | 29% |
| | without guided filter–quadratic | $R^2$ | linear | $R^2$ |
| gray matter | $28 - 0.025 * \text{age} - 0.0001 * \text{age}^2$ | 29% | $29 - 0.039 * \text{age}$ | 29% |
| white matter | $33 - 0.033 * \text{age} - 0.00004 * \text{age}^2$ | 17% | $33 - 0.038 * \text{age}$ | 17% |
| putamen | $35 - 0.027 * \text{age} - 0.0001 * \text{age}^2$ | 23% | $35 - 0.039 * \text{age}$ | 23% |
| pallidus | $38 - 0.037 * \text{age} - 0.00005 * \text{age}^2$ | 18% | $38 - 0.043 * \text{age}$ | 19% |

Table 4: Quadratic and linear regression results over the means of the considered tissue types for the experiment of age related statistics in Sec. 3. The regression results and $R^2$ values with guided filter are more similar to the results of previous studies (Ge et al., 2002), since a quadratic approximation delivers higher $R^2$ values. Without guided filter a more linear nature of the results can be observed since the second order coefficients are considerably lower and the $R^2$ values stay the same when a linear regression is performed.

The regression analysis of the curves presented in Fig. 3.5 delivers the equations shown in Tab. 4. For comparison, next to quadratic regression, we establish a linear regression to find the equations that fit the best. The fitting quality is described by the given $R^2$ value.

