# OpenReview forum: "Guided Filter Regularization for Improved Disentanglement of Shape and Appearance in Diffeomorphic Autoencoders"
_MIDL.io/2021/Conference — MIDL 2021_

### Official Review · AnonReviewer2 · 2021-03-08

**Confidence:** 4
**Preliminary Rating:** 3
**Recommendation:** Poster

**Summary:**

The paper presents an extension for the relatively recent mechanism of diffeomorphic/deformation auto-encoders that propose to disentangle shape and appearance of an image. The authors solve the issue of reliable/consistent mapping and disentaglement of shape and appearance by introducing a global template with a guided filter regularization of the appearance (the latter being the main innovation).

**Strengths:**

- novel method for use  in diffeomorphic/deformation auto-encoders (with the goal of separating shape and appearance, as each has separate decoders)
- main novelty is the added guided filter regularization of the appearance
- applied to a databased or T1 and T2 weighted MRI images

**Weaknesses:**

- this is all done on 2D brain slices of relatively small size (173x211). It is quite unclear that the proposed method would be able to handle the added complexity of 3D images (as MRI data should be handled in 3D).
- Adding a global template is not novel and the added guided filter for appearance regularization adds iterative novelty
- the T1 and T2 weighted images are not calibrated and thus one should not compute any direct intensity based statistics (such as age correlations). From a biological point of view, it not clear why the additional of the guided filter should provide more plausible/better results.
- in the dice results (from label maps), the proposed method may outperform the others (just by 2% dice), yet the dice scores are overall quite low for all methods.

**Deanonymize Review:**

no

**Justification Of The Preliminary Rating:**

This is clearly a borderline paper. The topic is interesting and relatively new (diffeomorphic/deformable auto encoders), yet the added novelty is quite limited. The evaluation is also limited, largely with visual results, as the quantitative results have a questionable application value.

**Paper Type:**

methodological development

**Special Issue:**

no

---

> ### Author Response · Authors · 2021-03-17
> **Answer to AnonReviewer2**
>
> Thank you for your helpful review. Please, see the general answer to all reviewers concerning the questions about the data dimensionality.
>
> We would like to address the reviewer's concern about the experiment for age-related statistics. As mentioned in the work,
> we do not have any quantitative MRI sequences at hand, thus no reliable analysis of the underlying structural changes using MTRs (magnetization transfer ratios) is possible. Even though we ensure comparable image quality by choosing images with the same acquisition parameters, we agree that calibrated MRIs would have contributed to more precise statistics. However, this experiment underlines several properties of the proposed method. First, due to the GF regularization, small adjacent structures with a strong difference in their intensity values (gray and white brain tissue structures here), can be differentiated more easily. This is due to the fact, that when no GF is used, the appearance map causes some blurriness which can be especially disruptive for adjacent small structures.Contrary to that, using GF regularization causes less blurriness and thus leads to better discriminability of the structures. Secondly, it is not explicitly the guided filter that ensures more plausible results. It is moreover the improved registration (as shown in previous experiments) and thus the better label segmentation which leads to less mixing of different intensities in one structure, that is largely responsible for the better results achieved by our approach.
>
> Furthermore, the reviewer mentions that the presented Dice results seem to be bad overall.
> We do not fully agree with this statement, as works on similar data also report similar Dice coefficients. The state-of-the-art deep learning approach for registration VoxelMorph [1] yields comparable results (see Tab. 1 in [1] for brain structures). Also this work shows that registration results of different algorithms often differ by just 0.5\% Dice coefficient.  We would like to emphasize that our approach performs a fully unsupervised group-wise registration and that the achievable Dice coefficient depends on the quality and diversity of the underlying labels.
>
>
> [1] Balakrishnan, G. et al. VoxelMorph: A Learning Framework for Deformable Medical Image Registration. IEEE Transactions on Medical Imaging. 2019.

---

### Official Review · AnonReviewer4 · 2021-03-08

**Confidence:** 4
**Preliminary Rating:** 3

**Summary:**

The authors propose a new system to disentangle shape and appearance in diffeomorphic autoencoders. A guided filter smoothing operation allows for multi-modal image modeling and crisper registration results as well as automated population analysis. The experimental results are promising and convincing and were run on a data set of roughly 600 images.

**Strengths:**

The submission is well written, clear and easy to read.

The new system models the shape and appearance of medical images as spatial and intensity offsets from a global template. This allows for robust and accurate group-wise registration of the input images. Its integration with appearance regularization, guided filtering, adds novelty and further improves the system's performance. The resulting template is more representative and is sharper.

The experimental results are appealing. The age-related statistics experiments were clearly described and a good example of the strength and usability of the newly proposed system.

**Weaknesses:**

How is it possible to use "around" 600 input images (Sec 3.1)? The authors should know exactly how many input datasets were used. Was this an adult dataset or a pediatric one?

It is only very briefly mentioned in the experimental section, on page 4, that the system described is for 2D images. I wish it was more clearly stated / mentioned earlier on.

**Deanonymize Review:**

no

**Detailed Comments:**

Please use a comma after "however" (which appears a lot) to make sentences easier to parse.
constrains --> constraints
preciser --> more precise
two atlase --> two atlases

**Justification Of The Preliminary Rating:**

The submission is clearly written, provides a methodological contribution to the field, and demonstrates the importance and usability of the new system. The preliminary experimental validation, a well-chosen experimental example using atlas-based segmentation, is based on a small data set (30 subjects, 10 labels).

**Paper Type:**

methodological development

**Special Issue:**

no

---

> ### Author Response · Authors · 2021-03-17
> **Answer to AnonReviewer4**
>
> We would like to thank for this thoughtful and positive review. Please, see the general answer to all reviewers concerning the questions about the data dimensionality.
> Concerning the questions about the dataset, the reviewer points out the imprecision of the number of images used. We used  577 images in our experiments and mentioned this in the revised version of the work. The dataset consists of adult subjects, which should be visible from the cited work of Hammers et al. [1] but is now additionally clarified.
>
> Further minor grammatical issues mentioned by the reviewer were also adapted in the revised version of the work.
>
> [1] Hammers A et al. "Three-dimensional maximum probability atlas of the human brain, with particular reference to the temporal lobe", Hum Brain Mapp 2003

---

### Official Review · AnonReviewer1 · 2021-03-08

**Confidence:** 4
**Preliminary Rating:** 3
**Recommendation:** Poster
**Final Rating:** 3

**Summary:**

The authors aim to address the task of modeling both image appearance and anatomical shape, and learning these two properties in a such a way that both are independent from each other (disentanglement of shape and appearance). To perform this task, they use the incorporate a global template and devise a network to model appearance and deformation (geometric) changes from this template as two output heads from a single, joint latent space. The authors also utilize a guided filter (from He et al 2013) at the end of the appearance decoder as an edge preserving smoothing filter. Experimental results using public brain MRI data demonstrates limited discernible qualitative improvements in reconstruction (as noted by the authors) but shows marginally improved quantitative improvements in terms of reconstruction quality and registration of ground-truth structures of interest.

**Strengths:**

•	Including a global template image as part of the method is nice because it introduces a global appearance/shape constraint into the model and offers some level of interpretability into the results.
•	Compare to previously reported methods (Shu et al, and Bone, et al.) as baseline
•	Strong 4-fold cross validation experiments provide results covering the entire test set.
•	Ablation experiments to test different training inputs is good: single modality T1 and T2 MRI and mixed modality training.
•	Ablation experiments show the effect of using guided filtering or not.
•	The included application study (brain appearance correlation with age) is nice to demonstrate practical application of the method.
•	Well written manuscript.


**Weaknesses:**

•	It is unclear from the text if this approach is applied to 2D images and if these 2D images are from a single location in the template or applies to multiple slices from the full 3D volume.
•	The application study does not compare to baseline methods, only the proposed model with or without guided filtering.
•	The improvements provided by the use of guided filtering are not convincing (results show in Tables 1 and 3 are not consistent).
•	Some network training parameters and implementation details are missing.


**Deanonymize Review:**

no

**Detailed Comments:**

The formulation to disentangle shape from appearance in a deep learning setting is very interesting. While results do not demonstrate substantial improvements in image reconstruction quality, the methodological innovation is of interest. The use of guided filtering is also good, but results are not consistent in terms of image reconstruction quality or registration accuracy improvement.


The paper lacks some details with respect to training parameter: What optimization method and parameters were used? How many training epochs were used? With what software is the network implement and what computational resources? Number of trainable parameters in each model (number of features in the conv layers)?


Equation 2: What values were used for alpha and beta?


Table 1: It would be interesting to see the variance in test results by including standard deviation for all results.


Sec. 2.3: Please clarify if this network operates on 2D data or 3D volumes in the text. It appear that this is model using 2D operations. But, does this approach model a single 2D image slice from the middle of the volume, or from all 2D slices in the 3D volume? Please add details to the implementation to reduce ambiguity.


Sec. 3.1: Please cite what registration method was used to register all images to the atlas space.


Sec. 3.5: Did you also test the baseline methods for the application study? If so, how does their performance compare to your proposed method?


Table 3: It is unclear from the caption what this table is showing. What is meant by “specificity” and “generalization” here?



Minor Comments:

Fig 2: It would be helpful to reader if you could label the different approaches in the figure and define them in the caption so that the reader could understand the figure as in a standalone manner, e.g. “diffeo” refers to the “Diffeomorphic autoencoder (Bone et al 2019)”, etc.


Grammatical:
Table 2: “atlase” -> “atlas”


**Final Rating Justification:**

I thank the authors for their revisions. Given the paper’s aim to address disentanglement of shape and appearance, I think the proposed method would be of interest to the MIDL community and would stimulate good conversations around this topic.

**Justification Of The Preliminary Rating:**

The formulation to disentangle shape from appearance in a deep learning setting is very interesting. While results do not demonstrate substantial improvements in image reconstruction quality, the methodological innovation could be of interest to the community.

**Paper Type:**

methodological development

**Questions To Address In The Rebuttal:**

Please provide additional information regarding the network implementation and training details.

Please provide parameter choices for the components of the loss function.

Sec. 3.5: Did you also test the baseline methods for the application study? If so, how does their performance compare to your proposed method?



**Special Issue:**

no

---

> ### Author Response · Authors · 2021-03-17
> **Answer to AnonReviewer1**
>
> We thank the reviewer for their helpful and detailed review. Please, see the general answer to all reviewers concerning the questions about the data dimensionality. Also, we are thankful for pointing out the missing architectural details and parameter descriptions. To clarify those, we now have made our source code publicly available and will add a reference to it in the final version of the work (https://github.com/hristina-uzunova/GF_DAE). The loss function parameters (for the shown experiments we set both $\alpha$ and $\beta$ to 10) and all other preferred settings are also available in the source code.
>
> Concerning the results from Sec. 3.5., we aimed to point out the importance of using the guided filter regularization against no regularization, and thus small structures that would typically get blurred appearance due to the lack of regularization were picked for this experiment. A comparison to the deformable auto-encoder is not possible in this case, since using a fixed atlas for all images is not enabled by the nature of the method. A comparison to the diffeomorphic approach would be, however, interesting for future works.
>
> A major concern of the reviewer is the fact that the quantitative reconstruction results are not improved by the presented approach. We would like to mention that the main focus of the presented method is to improve disentanglement of appearance and shape, rather than improve image reconstruction. Indeed, it is not expected that a better image reconstruction is achieved by adding regularization, since unregularized methods can compensate bad displacements by adjusting the appearance and vice versa. E.g. enlarged ventricles are not modeled as displacements but rather by adding pixels of suitable intensity to the additional map around the ventricles in the template (as shown in Fig. 5). Since all shown methods optimize toward the same objective, namely reconstruct input images by using a pixel-wise loss, the reconstruction results of all methods are comparable. However, we do not agree that the results from Tab. 1 and Tab. 3 are inconsistent, since Tab. 1 shows the registration results and Tab. 3 shows results for the quality of reconstruction of unseen samples and generation of new samples. Showing comparable results for the reconstruction but improved results for the registration task, actually leads to the conclusion that the GF approach is able to disentangle the displacements needed for image registration from the generated appearance and still be able to achieve the same image reconstruction results. The displacement fields of the other approaches are thus compensated by the appearances.
>
> Further remarks, especially concerning the quality of table and figure captions will be taken into account in the final version in case of acceptance. Also the standard deviations missing in Tab. 1 are included in the revised version as follows:
>
> ________________________________________________________________
> |0.76(0.08)|0.69(0.11)|0.74(0.10)  |0.76(0.10)
> ________________________________________________________________
> |0.64(0.11)|0.63(0.11)|0.67(0.09)  |0.69(0.09)

---

### Official Review · AnonReviewer3 · 2021-03-09

**Confidence:** 5
**Preliminary Rating:** 2
**Recommendation:** Poster

**Summary:**

This paper proposes a deep-learning based (pseudo) metamorphic atlas construction for multi-modal images. Following the Grenander’s template orbit theory, authors propose to learn a “global” template of the training population (given an initial estimate) and a (pseudo) metamorphic deformation for each image, disentangling shape and appearance variations.

**Strengths:**

-	Authors propose to use a guided filter for appearance regularization which seems to facilitate disentanglement between shape and appearance in multi-modal data-sets
-	Some results are interesting and well presented


**Weaknesses:**

-	No adequate comparison with Bone et al. 2020b
-	Lack of mathematical and statistical details about the proposed method
-	It's not clear whether and how the method can handle topological variations
-	Some results and figures are not well explained


**Deanonymize Review:**

no

**Justification Of The Preliminary Rating:**

This paper tackles an interesting and important problem in medical imaging which is the registration and atlas construction of multi-modal images using a deep learning strategy. In particular, they propose to use a (pseudo) metamorphic autoencoder, as in Bone et al. 2020b, regularizing the estimate of the appearance variation with a guided filter. This last idea is quite interesting since it helps disentangling shape and appearance variations. However, more details and discussion is needed to help the reader better understand the paper.

-	First of all, authors do not sufficiently compare their model with the work of Bone et al. 2020b. It seems that the only difference is the addition of the guided filter for appearance regularization. Is it the case?
-	Furthermore, authors do not sufficiently explain the statistical/mathematical details of their model. Do the authors use the same model as in Bone et al. 2020b? Is the template estimated as a Fréchet mean?  Do they also use RKHS for the shape and appearance fields? Is the input of their network an image or the difference between the template and the input image?
-	Authors should also speak about the atlas construction literature (estimating both a template and the subject-specific deformations) and make a link with the existing metamorphic/morphing algorithms (see metamorphosis of Trouvé et al. in computational anatomy, for instance)
-	Looking at Fig.4 it seems that the proposed model can take into account three different variations: shape, appearance and topology. Due to the use of the guided filter, appearance variations cannot modify the shape and topology of the guiding template, namely add new edges. This means that if a sample has a different topology (i.e. a tumor), this new structure must be present in the estimated global template to obtain a correct matching. Is it correct? If it’s the case, authors should better explain this point in the paper and probably change Fig.4 (now it’s definitely not clear). I also suggest that they add pathological images (i.e. with tumors) in their training dataset. The sentence -“Structures available only in a subset of images, are still generated in the template and their presence is controlled by the additional map (Figure 4).”- is not so clear. Furthermore, I guess that the disentanglement between topological and appearance changes also depends on some hyperparameter. Authors should better discuss this point.
-	It would also be interesting to show the evolution of the estimated template similarly to Fig. 5 for shape and appearance variations (which, by the way, is a very nice figure). Again, if I correctly understood the paper, topological changes should only appear in the evolution of the template.


**Paper Type:**

methodological development

**Special Issue:**

no

---

> ### Author Response · Authors · 2021-03-17
> **Answer to AnonReviewer3**
>
> We would like to thank the reviewer for their helpful and interesting comments and suggestions.
>
> As mentioned in the introduction and pointed out by the reviewer, our approach without GF shares some similarities with the very recent paper of Bone et al. [1]. Both approaches model images as additional intensity offsets and diffeomorphic shape variations of a template image. While Bone et al. apply their approach in the context of modeling pathological structures, we consider modeling of multimodal data for group-wise registration and template generation.
>
> The reviewer would like to see a more detailed comparison of the two approaches. From an experimental perspective, a fair comparison is impossible because a public implementation of [1] is not available and the paper lacks some implementation details.
>
> From a theoretical point of view, this comparison is very interesting since Bone et al. approach the problem from the perspective of geodesic image matching and geometric metamorphosis [2,3,4]. Accordingly, they consider a non-parametric setting where both velocity field and intensity offset are embedded in an RKHS. Explicit filter layers are used in the network to ensure the properties of each RKHS and $X-T$ is used as network input to enforce that $T$ is the Fr´echet mean of the training images.
> In our approach, we consider the task as a regularized optimization problem in a parametric setting, such that $v(z_v)$ and $\Delta(z_A)$ are continuous differentiable functions with respect to the parameters, and regularization terms enforce plausibility and well-posedness of the solution.
> A diffusive regularization term (approximated by Gaussian smoothing in [1] (see [5], Chap. 11)) ensures the spatial smoothness of the velocity field, which is necessary to obtain diffeomorphic transformations. The KL divergence ensures that $z_v, z_A\sim N(0,I)$, together with tanh activations and weight decay (L2 penalty) of the network parameters, this yields expectations $E[v]=0$ and $E[\Delta]=0$ as the optimization process approaches the global optimum. Thus, at least the approach without GF can be interpreted as computing a Fr´echet mean of the training data set for an image distance $d(X,X’)=||v-v’||+||\Delta - \Delta’||$.
> Due to the different underlying concepts, our implementation differs substantially from [1] (e.g. our input is $X$ and not $X-T$), thus an experimental comparison is interesting as well.
> Main differences between both concepts will be described in a revised version, however, a thorough analysis with experimental substantiation is currently infeasible.
>
> A further question of the reviewer is related to the representation of topological variations in the template. As mentioned above, we consider our approach in the context of multimodal images, where some structures might be visible in only one modality. Without GF and assuming a L2 distance to measure image similarity, the template will represent these structures by the average of the intensities weighted according to the prevalence in the training set, where as structures not visible/represented in the template can be generated by the appearance map. This is not the case with GF as only structures represented in the template can be emphasized or de-emphasized. Thus, rather than learning a simple intensity average, the network learns to represent relevant structures in the training set by edges in the template image.
> The relevance of image structures is related to their contribution to the overall loss, i.e. their size and frequency in the training set. Further, the size of the latent space and capacity of the decoder determine the "level of relevance" that the autoencoder can reconstruct. Fig. 4 shows that this also applies in a monomodal setting to anatomical variations that are only present in a subset of the training images. We will update the figure caption to be more comprehensive. In the case of pathologies (e.g. tumors), as mentioned by the reviewer, their variability and low frequency in the training set will likely prevent the network from learning a representation in the template. Therefore, this approach is  not intended for the reconstruct pathologies.
> The evolution of the estimated template image during the training process would certainly be an interesting  presentation but due to page limitations, we will consider including a corresponding figure in the appendix.
>
> [1] Bône, Alexandre, et al. "Learning joint shape and appearance representations with metamorphic auto-encoders." IMICCAI, 2020.
>
> [2] Grenander, Ulf, and Michael I. Miller. "Computational anatomy: An emerging discipline." Quarterly of applied mathematics, 1998
>
> [3] Trouvé, Alain, and Laurent Younes. "Metamorphoses through lie group action." Foundations of computational mathematics, 2005
>
> [4] Niethammer, Marc, et al. "Geometric metamorphosis." MICCAI,  2011.
>
> [5] Modersitzki, Jan. Numerical methods for image registration. Oxford University Press on Demand, 2004.

---

### Author Response · Authors · 2021-03-17
**Answer to all reviewers concerning the image dimension**

First, we would like to thank the reviewers for their thoughtful reviews and the overall appreciation of the proposed method.
Since many reviewers had questions concerning the dimension of the used images, we would like to clarify this issue.

As we briefly mention in the paper, we use the approach only for 2D slices, extracting one slice  per 3D volume at corresponding locations. An extension to large 3D volumes ($256^3$ voxels and larger) is essentially constrained by hardware limitations (i.e. GPU memory). We are  aware of the 3D nature of brain MRIs, and currently explore patch-based techniques [1] to circumvent those limitations. However, we consider the application on 2D images as an adequate proof-of-concept of the introduced novelties. This will be clarified in a revised version of the paper.

[1] Uzunova H, et al.: “Memory-efficient GAN-based domain translation of high resolution 3D medical images”, CMIG 2020

---

### Meta-Review · Area_Chair1 · 2021-03-28

**Recommendation:** Accept (Poster)

**Metareview:**

This is a borderline paper. The reviewers appreciate the method, which is interesting and of high current interest, but there are also concerns regarding the experimental validation, which is promising but limited.

**Paper Type:**

methodological development

---

### Decision · Program_Chairs · 2021-03-31

Accept